# Expansion and Shrinkage of Localization for Weakly-Supervised Semantic Segmentation

**Jinlong Li**[1,2,*]     **Zequn Jie**[2,*]     **Xu Wang**[1,†]     **Xiaolin Wei**[2]     **Lin Ma**[2,†]
[1] College of Computer Science and Software Engineering, Shenzhen University, China
[2] Meituan Inc.
jinlong.szu@gmail.com, zequn.nus@gmail.com, wangxu@szu.edu.cn,
weixiaolin02@meituan.com, forest.linma@gmail.com

## Abstract

Generating precise class-aware pseudo ground-truths, *a.k.a*, class activation maps (CAMs), is essential for Weakly-Supervised Semantic Segmentation. The original CAM method usually produces incomplete and inaccurate localization maps. To tackle with this issue, this paper proposes an Expansion and Shrinkage scheme based on the offset learning in the deformable convolution, to sequentially improve the **recall** and **precision** of the located object in the two respective stages. In the Expansion stage, an offset learning branch in a deformable convolution layer, referred to as "expansion sampler", seeks to sample increasingly less discriminative object regions, driven by an inverse supervision signal that maximizes image-level classification loss. The located more complete object region in the Expansion stage is then gradually narrowed down to the final object region during the Shrinkage stage. In the Shrinkage stage, the offset learning branch of another deformable convolution layer, referred to as "shrinkage sampler", is introduced to exclude the false positive background regions attended in the Expansion stage to improve the precision of the localization maps. We conduct various experiments on PASCAL VOC 2012 and MS COCO 2014 to well demonstrate the superiority of our method over other state-of-the-art methods for Weakly-Supervised Semantic Segmentation. The code is available at https://github.com/TyroneLi/ESOL_WSSS.

## 1 Introduction

Image Semantic Segmentation is the task of pixel-level semantic label allocation for recognizing objects in an image. The development of Deep Neural Networks (DNNs) has promoted the rapid development of the semantic segmentation task [7, 20, 64] in recent years. However, training such a Fully-Supervised Semantic Segmentation model requires large numbers of pixel-wise annotations. Preparing such a segmentation dataset needs considerable human-labor and resources. Recently, researchers have studied Weakly-Supervised Semantic Segmentation (WSSS) methods to alleviate the issue of high dependence on accurate pixel-level human annotations for training semantic segmentation models under cheap supervision. Weak supervision takes the forms of image-level [2, 21, 52, 54, 55], point-level [3], scribbles [36, 50] or bounding box [11, 28, 32, 47]. In this paper, we focus on WSSS method based on image-level labels only, because it is the cheapest and most popular option of weak supervision annotation which only provides information on the existence of the target object categories.

---

[*]Equal contributions. [†]Corresponding authors.

Most WSSS methods utilize class labels [2, 21, 52, 54, 55] to generate pseudo ground-truths for training a segmentation model obtained from a trained classification network with CAM [65] or Grad-CAM [44] method. However, image-level labels can not provide specific object position and boundary information for supervising the network training, resulting in that these localization maps identify only local regions of a target object that are the most discriminative ones for the classification prediction. Therefore, with the incomplete and inaccurate pseudo ground-truths, training a Fully-Supervised Semantic Segmentation network to reach a decent segmentation performance is challenging. Existing WSSS methods usually attempt to gradually seek out more less discriminative object regions starting from the very small and local discriminative regions [1, 2, 6]. Differ from the existing works, in this paper, we attack the partial localization issue of the CAM method with a novel deformable transformation operation. We empirically observe that the classification models can re-discover more discriminative regions when we fix a trained classifier and equip the network with more "sampling" freedom to attend to other less discriminative features. This inspires us to explore a proper way to improve the quality of the initial localization maps via a new training pipeline following a Divide-and-Conquer manner, **Expansion and Shrinkage**, shown in Figure 1.

The **Expansion** stage aims to recover the entire object as much as possible, by sampling the exterior object regions beyond the most discriminative ones, to improve the **recall** of the located object regions. We introduce a deformable convolution layer after the image-level classification backbone, whose offset learning branch serves as a sampler seeking for sampling increasingly less discriminative object regions, driven by an inverse image-level supervision signal. We call this newly embedded deformable convolution layer "expansion sampler" (ES). During the inverse optimization process, the backbone network is frozen to provide fixed pixel-wise features obtained

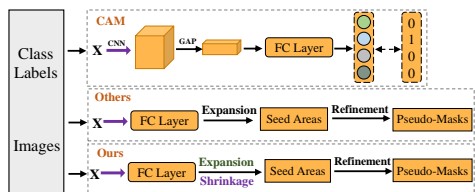

Figure 1: The pipeline comparison for training a WSSS model. Our contribution is to improve the quality of initial localization maps generated by the classification network with a new "Expanding first then Shrinking" scheme.

in the image-level classification training to be sampled by the offset learning branch in the ES. In this way, the inverse supervision target solely enforces the offset learning in the ES branch to optimize its sampling strategy to gradually attend to the less discriminative regions, given that the pixel-wise features cannot be changed. Hence, the image-level loss maximization allows the network to pay more attention to the less discriminative regions via deformation transformation achieved by ES in the inverse optimization, which are easily ignored during the normal image-level classification task.

Having obtained the high-**recall** object region after the **Expansion** stage, we then propose a **Shrinkage** stage to exclude the false positive regions and thus further enhance the **precision** of the located object regions. The Shrinkage stage remains the same network architecture as in the Expansion stage except that an extra deformable convolution layer, referred as "shrinkage sampler" (SS), is introduced to narrow down the object region from the high-**recall** one. However, we observe a feature activation bias issue, *i.e.*, the initial most discriminative parts are more highlighted in the feature map after the ES in the Expansion stage while the newly attended regions have much weaker feature activation. Such activation bias serves as prior knowledge which encourages the later shrinkage to converge to the same discriminative parts as the initial highlighted regions in the original CAM. To alleviate such an issue, we introduce a feature clipping strategy after the ES in the Expansion stage of training to normalize pixel-wise feature values, allowing each target pixel to be activated evenly and have a relatively fair chance to be selected by the SS in the Shrinkage stage. Similarly, in the training of Shrinkage stage, all the layers before the SS are fixed to provide stable pixel-wise features and only the offset learning branch in the SS is updated to sample the true positive pixels, optimized by the standard image-level classification supervision.

The main contributions of this paper are summarized as follows. First, this paper proposes an Expansion and Shrinkage scheme to sequentially improve the **recall** and **precision** of the located object in the two respective stages, leading to high-quality CAM which can be used as strong pseudo ground-truth masks for WSSS. Second, both the Expansion and Shrinkage stages are realized by carefully applying deformable convolution combined with two contrary training signals. To avoid the repeated convergence to the initial discriminative parts, a feature clipping method is applied to alleviate the activation bias of these regions. Third, our approach significantly improves the quality of

the initial localization maps, exhibiting a superior performance on the PASCAL VOC 2012 and MS COCO 2014 datasets for WSSS.

## 2 Related Work

### 2.1 Weakly-Supervised Semantic Segmentation

Weakly-Supervised Semantic Segmentation pipeline [21, 27] with image-level labels only mostly consists of two steps: pseudo ground-truths generation and segmentation model training [1, 2, 8, 23, 26, 33, 34, 53–55]. Erasure methods [19, 46, 54] applied various iterative erasing strategies to prevent the classification network from focusing only on the most discriminative parts of objects by feeding the erased image or feature maps to the model. MDC [55], layerCAM [24] and FickleNet [30] aggregated different contexts of a target object by considering multiple attribution maps from different dilated convolutions or the different layers of the DCNNs. Some works utilized diverse image contexts to explore cross-image semantic similarities and differences [16, 49]. CONTA [62] analyzed the causalities among images, contexts and class labels and used intervention to remove the confounding bias in the classification network. Recently, Anti-Adv [31] utilized an anti-adversarial manipulation method to expand the most discriminative regions in the initial CAMs to other non-discriminative regions. RIB [29] used the information bottleneck principle to interpret the partial localization issue of a trained classifier and remove the last double-sided saturation activation layer to alleviate this phenomenon. However, the generated location maps obtained by the classifier cannot reveal the entire object areas with accurate boundaries, the initial CAM seeds obtained using the methods above were further refined by a subsequent refinement network [1, 2, 8]. In this paper, we also follow this pipeline and our contribution is to propose a new training pipeline to generate high-quality localization maps. Different from MDC [55] using multi-dilated convolutions to combine multiple contexts for better feature mining, our method utilizes deformation transformation for less discriminative feature discovery.

### 2.2 Deformation Modeling

We refer to deformation modeling as learning geometric transformations in 2D image space without regarding to 3D. One popular way to attack deformation modeling is to craft certain geometric invariances into networks. However, to achieve this usually needs specific designs for certain kinds of deformation, such as offset shifts, rotation, reflection and scaling [4, 10, 13, 25, 45, 56]. Another line of this work on deformation modeling task learns to recompose data by either semi-parameterized or completely free-form sampling in image space. STN [22] learnt 2D affine transformations to construct feature alignment. Deformable Convolutions [12, 66] applied learnable offset shifts for better feature learning in free-from transformations. In WSSS community, applying deformation modeling is still less explored. In this paper, we utilize deformation transformation to act as a feature "sampling" to re-discover other non-discriminative regions, instead of better feature representation learning.

## 3 Proposed Method

Weakly-Supervised Semantic Segmentation methods use given class labels to produce pixel-level localization maps from a classification model using CAM [65] or Grad-CAM [44]. We first give some fundamental introduction to localization maps generation with CAM [65] in Section 3.1. Then, we present the whole framework of our method, **Expansion and Shrinkage with Offset Learning (ESOL)**, to obtain high-quality localization maps covering more complete and accurate target object regions in Section 3.2 and Section 3.3, respectively. We then explain how we train the final semantic segmentation model with the generated localization maps in Section 3.4.

### 3.1 Prerequisties

We first present the way to generate localization maps via the CAM [65]. A class activation map of a target object focuses on the regions of an image by a trained classification network for a specific category prediction. The CAM is based on a DCNN with a global average pooling (GAP) before its final classification layer, which is trained by a sigmoid cross-entropy loss function, formulated as follows:

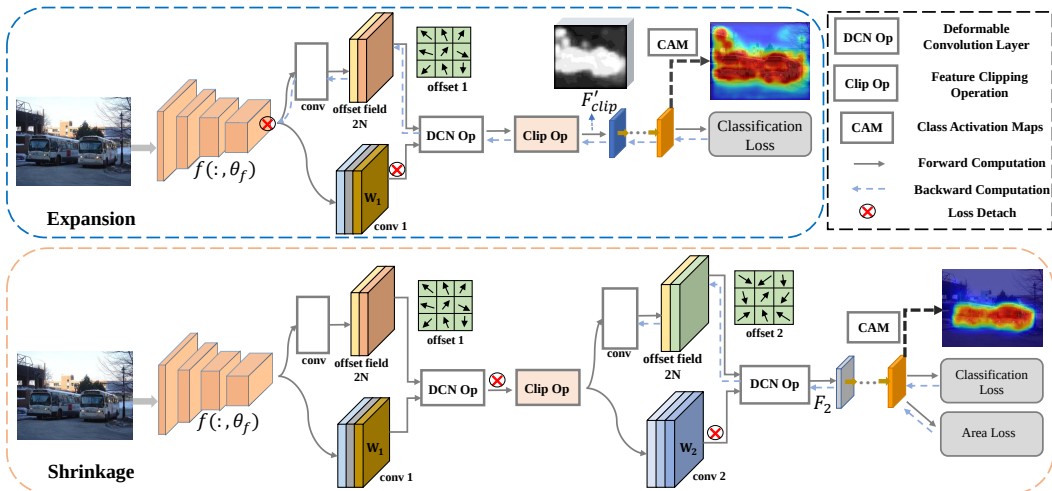

Figure 2: Our proposed Expansion and Shrinkage training pipeline. The **Expansion** scheme consists of a feature extractor $f(:, \theta_f)$, deformable convolution layer, hand-craft feature clipping operation and classifier layer. A loss maximization training is implemented to enable the offset learning in the ES to attend to other less-discriminative regions. For **Shrinkage** scheme, an extra deformable convolution layer is introduced to exclude the false positive regions with a loss minimization optimization, including a classification loss and an area loss function, respectively. The Expansion and Shrinkage are optimized via contrary training signals to sequentially improve the recall and precision of the initial localization maps.

$$\mathcal{L}(\hat{y}, y) = -\frac{1}{C} * \sum_i y[i] * \log((1 + e^{-\hat{y}[i]})^{-1}) + (1 - y[i]) * \log(\frac{e^{-\hat{y}[i]}}{1 + e^{-\hat{y}[i]}}), \quad (1)$$

where $C$ is the total number of training classes, $i$ represents the $i^{th}$ training sample in the mini-batch, $y[i]$ is the ground-truth label of $i^{th}$ class with the value of either 0 or 1 while $\hat{y}[i]$ is the model prediction. The localization maps is realized by considering the class-specific contribution of each channel of the last feature map, before the GAP layer, to the final classification prediction. Given a trained classifier network parameterized by $\theta = \{\theta_f, \mathbf{w}\}$ where $f(:, \theta_f)$ is the feature extractor, and $\mathbf{w}$ denotes the weight of the final classification layer. For some class $c$, the localization map is then computed from an input image x as follows:

$$\text{CAM}(\text{x}; \mathbf{w}) = \frac{\mathbf{w}_c^T f(\text{x}; \theta_f)}{\max \mathbf{w}_c^T f(\text{x}; \theta_f)}, \quad (2)$$

where $\max(\cdot)$ is the maximization over the spatial locations for normalization. The above method can only locate the most discriminative regions and fails in locating other less-discriminative regions that are semantically meaningful as well. In the following work, we elaborate our method by presenting a new training pipeline to capture high-quality object localization maps.

## 3.2 Expansion

As mentioned above, the localization maps generated by a commonly trained classifier usually struggle with the partial localization issue of the target objects, since the image-level labels cannot provide detailed position or boundary information. To alleviate this issue, we devise a new training pipeline, Expansion, to firstly recover the entire object regions as much as possible so as to improve the recall of the located object regions. Then, we further introduce another training scheme, Shrinkage, to exclude false positive regions, *e.g.,* background regions, to enhance the precision of the located regions.

A commonly trained classifier usually considers only the local regions that make the most contributions to final classification prediction, resulting in incomplete and inaccurate pseudo labels. Differ from other works, we first enforce the network to seek out the entire target object regions via our proposed Expansion scheme with an offset learning branch in a deformable convolution layer, as shown at the top of Figure 2.

A trained classifier is utilized to prepare our Expansion scheme firstly providing the regular convolutional weights in a deformable convolution layer, that can capture the most discriminative feature

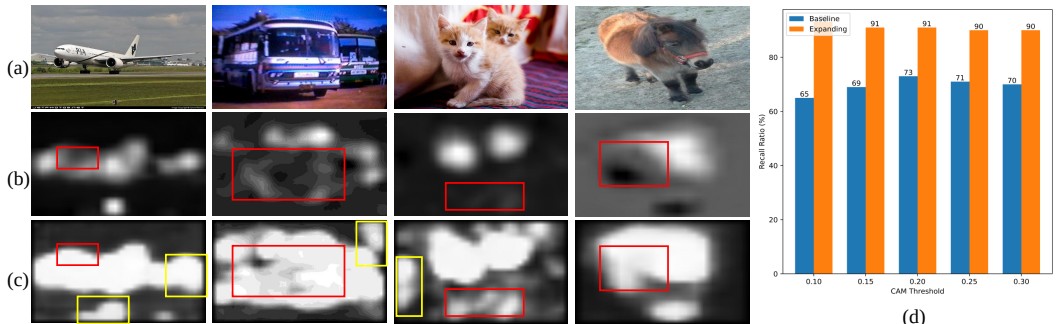

Figure 3: Feature visualizations for well demonstration of the proposed Expansion scheme. (a) Examples of input images. (b) Examples of feature visualization of $F_1$ from trained classifier. (c) Examples of feature visualization of $F'_{clip}$ from our Expansion stage. (d) Plot of Recall values of pre-trained classifier *v.s.* our Expansion stage to demonstrate the generated high-recall results, in terms of the foreground object Recall. Red boxes point out the difference between the trained classifier and Expansion while yellow boxes point out the negative positive regions (*e.g.,* background).

activation, *e.g., conv_*1 or *conv_*2. Given an input image x, the classification network $\theta = \{\theta_f, \mathrm{w}\}$ tends to locate the coarse object regions. For a specific convolution layer, *conv_*1 with trainable weights $\mathrm{w}_1$, the output feature maps $F_1$ is computed for each location $p_0$ as follows:

$$F_1(p_0) = \sum_{\mathrm{p_n} \in \mathrm{R}} \mathrm{w}_1(\mathrm{p_n}) \cdot \mathrm{x}(\mathrm{p}_0 + \mathrm{p_n}), \tag{3}$$

where R defines a specific kernel size with a specific dilation size (*e.g.,* 1) and $\mathrm{p_n}$ enumerates the locations in R. For Expansion scheme, a embedded deformable convolution layer is introduced after the feature extractor $f(:, \theta_f)$. The new feature maps $F'_1$ is then computed as:

$$F'_1(p_0) = \sum_{\mathrm{p_n} \in \mathrm{R}} \mathrm{w}_1(\mathrm{p_n}) \cdot \mathrm{x}(\mathrm{p}_0 + \mathrm{p_n} + \Delta \mathrm{p}_{1n}), \tag{4}$$

where the regular grid R is augmented with learnable offset fields $\{\Delta p_{1n} | n = 1, ..., N\}$, $N$ denotes the number of offset points, *e.g.,* $1 \times 1$ or $3 \times 3$. The learned $\Delta p_{1n}$ obtained from the offset learning branch in the deformable convolution layer further acts as "expansion sampler" to sample the exterior object regions gradually beyond the most discriminative ones. This training scheme utilizes an image-level loss maximization to enforce the network seek for increasingly less-discriminative object regions, given unchanged pixel-wise features that is implemented by detaching the loss back-propagation to the backbone. This can be validated by our experimental results shown in Figure 3. Thus, the training loss function $\mathcal{L}_{\mathrm{expansion}}$ utilized in Expansion scheme becomes:

$$\mathcal{L}_{\mathrm{expansion}} = -\alpha \mathcal{L}(\hat{\mathrm{y}}, \mathrm{y}), \tag{5}$$

where $\alpha$ (denoting positive numbers) controls the loss weight for updating and $\mathcal{L}(\hat{y}, y)$ is the corresponding multi-label classification loss function mentioned in Eq 1.

### 3.3 Shrinkage

After the Expansion stage, though most possible target object regions are sampled, the noisy backgrounds are also included. This would inevitably cause imprecise localization maps generation, which results in poor quality pseudo ground-truths and hampers the final segmentation performance. To enhance the precision of such high-recall regions, Shrinkage scheme is proposed to exclude the false positive regions of the localization maps. The Shrinkage stage remains the same network architecture as in Expansion stage, except that an extra deformable convolution layer is implemented to narrow down the high-recall regions. Specifically, the model weights obtained from the Expansion are used to initialize the Shrinkage model and a loss minimization is adopted to train the network, including a multi-label classification loss and an area loss. Area regularization is adopted to constraint the size of the localization maps to ensure that the irrelevant backgrounds are excluded in the localization map $\mathcal{P}_k$, which is also studied in [58, 59]:

$$\mathcal{L}_{\text{shrinkage}} = \gamma \mathcal{L}(\hat{y}, y) + \mu \mathcal{L}_{\text{area}}, \qquad \mathcal{L}_{\text{area}} = \frac{1}{C} \sum_{c=1}^{C} \mathcal{S}_{c}, \tag{6}$$

where $\mathcal{S}_c = \frac{1}{HW} \sum_{h=1}^{H} \sum_{w=1}^{W} \mathcal{P}_k(h, w)$, $C$ is the total class category numbers of the dataset, $H$ and $W$ denotes the height and width of the localization maps, respectively.

We empirically observe that the feature activation bias issue is obvious, the initial most discriminative parts got more highlighted after ES in the Expansion stage while the newly attended ones have weaker feature activation. This activation bias would encourage the later Shrinkage to seek for the same discriminative parts as the initial activation parts, even more local. To address such an issue, a feature clipping strategy is then proposed after the ES in the Expansion stage to normalize pixel-wise features, providing relatively fair chances for the pixels to be selected by SS in the Shrinkage stage. The feature clipping strategy is formulated as follows:

$$F'_{clip}(\mathrm{x}_i) = \begin{cases} \beta \max(\mathrm{x}), & \mathrm{x}_i \geq \beta \max(\mathrm{x}) \\ \mathrm{x}_i, & \text{others} \end{cases}, \tag{7}$$

where $\mathrm{x}_i$ is the input feature maps over the spatial dimension, $\max(\cdot)$ is applied to obtain the maximal values along the spatial dimension as well.

We present examples of PASCAL VOC 2012 [14] training images for better demonstration shown in Figure 5. During the Expansion stage, the image-level loss maximization enforces the offset learning branch attend on other less-discriminative regions, indicating that the ES indeed serves as a sampler locating the entire object regions, thanks to the deformation transformation in the deformable convolution layer. For example, the body or legs of the cat or cow shown in Figure 3 is successfully activated by our Expansion scheme, and the overall recall of foregrounds is significantly improved among diverse hard-threshold settings. Then, the included noisy background regions are excluded by the SS sampler in the Shrinkage stage, gradually sampling the true positive foreground regions and finally providing high-quality pseudo labels.

### 3.4 Pseudo Ground-truth Generation

The final localization map $M$ is generated by our final Shrinkage trained model via CAM [65] method. Since a CAM is computed from high-level small size intermediate feature maps produced by a classification network, it need to be up-sampled to match the size of the original image. Thus, it tends to localize the most discriminative target object regions coarsely and cannot cover the entire regions with exact boundary. Many WSSS methods [5, 30, 51, 53, 62] produce pseudo ground-truths by extending their initial CAM seeds using another seed refinement methods [1, 2, 8]. Similarly, we obtain our final pseudo ground-truths using IRN [1], a state-of-the-art refinement method, to refine the coarse map $M$ for generating better segmenta-

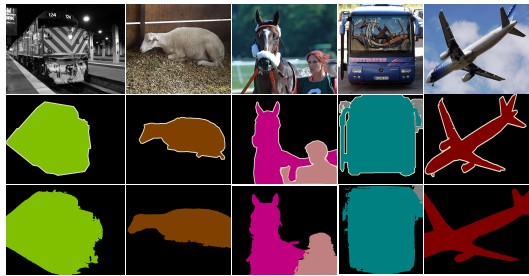

Figure 4: Examples of the final segmentation predictions on PASCAL VOC 2012 validation. The top row, middle row and bottom row denotes the input images, corresponding ground truth and our final segmentation model predictions, respectively.

tion model supervision. Then a semantic segmentation model is trained using the generated pseudo labels following Fully-Supervised manner and Figure 4 illustrates the final segmentation results on VOC2012 val set.

## 4 Experiments

### 4.1 Experimental Setup

**Dataset and Evaluation metric:** Experiments are conducted on two publicly available datasets, PASCAL VOC 2012 [14] and MS COCO 2014 [37]. The Pascal VOC 2012 dataset contains 20 foreground categories and the background. It has three sets, the training, validation, and test set, each

| Method | Refinement Method | PASCAL VOC | | | MS COCO | |
|---|---|---|---|---|---|---|
| | | Seed | CRF | Mask | Seed | Mask |
| PSA CVPR '18 [2] | PSA [2] | 48.0 | - | 61.0 | - | - |
| Mixup-CAM BMVC '20 [5] | | 50.1 | - | 61.9 | - | - |
| CDA ICCV '21 [48] | | 48.9 | 57.5 | 63.3 | - | - |
| SC-CAM CVPR '20 [6] | | 50.9 | 55.3 | 63.4 | - | - |
| ESOL (Ours) | | **53.6** | **61.4** | **66.4** | - | - |
| IRN CVPR '19 [1] | IRN [1] | 48.8 | 53.7 | 66.3 | 33.5‡ | 42.9‡ |
| MBMNet ACMMM '20 [39] | | 50.2 | - | 66.8 | - | - |
| BES ECCV '20 [8] | | 50.4 | - | 67.2 | - | - |
| CONTA NeurIPS '20 [62] | | 48.8 | - | 67.9 | 28.7† | 35.2† |
| CDA ICCV '21 [48] | | 50.8 | 58.4 | 67.7 | - | - |
| ESOL (Ours) | | **53.6** | **61.4** | **68.7** | **35.7‡** | **44.6‡** |

Table 1: Comparison of the initial localization maps (Seed), the seed with CRF (CRF), and the pseudo ground-truths mask (Mask) on PASCAL VOC 2012 and MS COCO 2014 training images, in terms of mIoU (%). † denotes the results reported by CONTA [62], and ‡ denotes the results obtained by us.

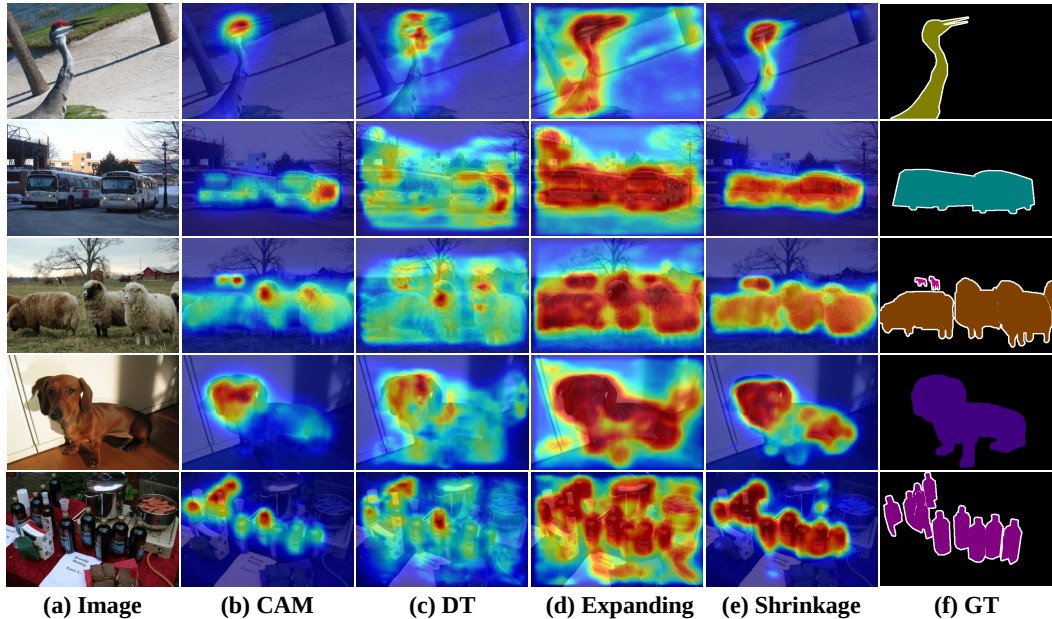

| **(a) Image** | **(b) CAM** | **(c) DT** | **(d) Expanding** | **(e) Shrinkage** | **(f) GT** |

Figure 5: Examples of localization maps on PASCAL VOC 2012 training images. (a) Input Images. (b) Original baseline CAMs. (c) DT denotes Expansion training without feature clipping strategy. (d) Expansion results. (e) Shrinkage results. (f) Ground Truth.

containing 1464, 1449 and 1456 images, respectively. Following most previous works [5, 30, 51, 53, 62], we also adopt the augmented training set [17] to yield totally 10582 training images. The MS COCO 2014 dataset has 80 foreground categories, including approximately 82K training images and 4K validation images. We evaluate our method on 1449 validation images and 1456 test images from the PASCAL VOC 2012 dataset and on 40504 validation images from the MS COCO 2014 datasets. The mean intersection-over-union (mIoU) [41] is used as the evaluation metric.

**Implementation Details:** We implement CAM [65] by following the procedure from Ahn *et al.* [1], implemented with the PyTorch framework [43] on 12G Nvidia XP Graphic Cards. We adopt the ResNet-50 [18] as backbone for the classification model. To prepare the regular convolutional weights for the deformable convolution layers, we opt to follow the common baseline settings to train a classifier for our method, except that we add two additional convolution layers ($3 \times 3$ or $1 \times 1$) that would be used to perform the deformable transformation in our proposed Expansion and Shrinkage, respectively. For Expansion stage, we train the network 6610 iterations for PASCAL VOC 2012 and 51730 iterations for MS COCO 2014. To carefully train the model with a loss maximization, we set a relatively small learning rate, 0.01 and 0.001 for PASCAL VOC 2012 and MS COCO 2014, while the controlling parameter $\alpha$ is set to 0.001. Noting that the loss gradients are not allowed to

back-propagate to the backbone, as we want to provide pixel-wise unchanged feature maps for ES and SS. For Shrinkage stage, the network is initialized from the Expansion model weights. The learning rate is set to be 0.1 and 0.02, and the training iteration is set to be 6610 and 51730 for PASCAL VOC 2012 and MS COCO 2014, respectively. The threshold value of the hand-craft feature clipping strategy is 0.15. The $\gamma$ and $\mu$ are both set to be 1.0. To generate reliable initial localization maps, the scale ratio of multi-scale CAM is $\{0.5, 1.0, 1.5, 2.0\}$. During testing, DenceCRF [28] is used as post-processing to refine the generated localization maps. For the final semantic segmentation, we use the PyTorch implementation of DeepLab-v2-ResNet101 provided by [42].

### 4.1.1   Quality of the initial localization maps and pseudo ground-truths

**PASCAL VOC 2012 dataset:** In Table 1, the mIoU values are used to estimate the quality of initial localization maps (denoted as seed) and pseudo ground-truths masks produced by our method, and other recent WSSS techniques. Following common practice [1, 2, 6, 53], we perform a range of hard-threshold values to distinguish the positive and negative regions in localization maps $M$ to determine the best initial seeds result. Our initial seeds improves 5.2% mIoU over the original CAM seeds (48.4% to 53.6%), a baseline for comparison, and outperform those simultaneously from the other methods. Noting that our initial seeds are superior to those of CDA [48] or SC-CAM [6], which applied a complicated context decoupling augmentation on the network training or adopted sub-category exploration to enhance the feature representation via an iterative clustering method.

### 4.2   Weakly-Supervised Semantic Segmentation

After obtaining the initial seeds based on CAM [65], we then apply the Conditional Random Field (CRF) [28] for pixel-level refinement on the results from the method proposed by SC-CAM [6],CDA [48], IRN [1], and our method in Table 1. We can observe that applying CRF improves all the initial seeds over 5% mIoU. When the seeds generated by our method is then post-

| Method | Backbone | mIoU (%) |
|---|---|---|
| ADL $_{TPAMI\ '20}$ [9] | VGG16 | 30.8 |
| CONTA $_{NeurIPS\ '20}$ [62] | ResNet50 | 33.4 |
| SEAM $_{CVPR\ '20}$ [53] | Wider-ResNet38 | 32.8 |
| IRN $_{CVPR\ '19}$ [1] | ResNet101 | 41.4 |
| ESOL (Ours) | ResNet101 | **42.6** |

Table 2: Comparison of semantic segmentation on MS COCO validation images.

processed with CRF, it obtains 7.8% mIoU better than the original CAM (53.6% to 61.4%), and consequently surpasses all the recent competitive methods. A seed refinement is applied to generate the pseudo ground-truth masks with other methods. For a fair comparison, we compute our pseudo ground-truths masks using both seed refinements, PSA [2] or IRN [1]. Table 1 illustrates the masks results from our method yield 66.4% mIoU with PSA [2] and 68.7% mIoU with IRN [1], respectively.

Figure 5 visualizes initial localization maps adopted different components or training phases for the PASCAL VOC 2012 dataset. And the visualizations demonstrate the effectiveness of the proposed Expansion and Shrinkage approach, sequentially balancing the recall and precision of the initial localization maps. More examples are shown in the Appendix.

### 4.2.1   Performance of the Weakly-Supervised Semantic Segmentation

**PASCAL VOC 2012 dataset:** Table 3 shows the segmentation performance (mIoU) on PASCAL VOC 2012 validation and test set. We illustrate the results predicted by our method and other recently proposed WSSS methods, which use either bounding box or image-level labels. All the segmentation results in Table 3 were implemented by a ResNet-based backbone [18]. Our proposed method achieves 69.9% and 69.3% mIoU values for the validation and test set on the PASCAL VOC 2012 dataset, outperforming all the WSSS methods that utilize image-level class labels only.

In particular, our method surpasses CONTA [62], a state-of-the-art WSSS competitors recently, obtaining 66.1% mIoU. CONTA adopted SEAM [53], which is applied with WiderResNet-based [57] backbone that is known to be more powerful than IRN [1] based on ResNet-based. When it was implemented with IRN [1] for a fair comparison with our method, its segmentation performance got only 63.5%, which is inferior to ours by 6.4% mIoU.

Recently, saliency map cues are introduced to supervise the network training for better localization performance, since the offline saliency maps provide detail foreground boundary priors. In Table 4, we also compare our method with other methods using extra saliency maps. We combine our final pseudo ground-truths masks with saliency maps produced by Li *et al.* [35] or Liu *et al.* [38]. We can see that

| Method | val | test |
|---|---|---|
| Supervision: Bounding box labels | | |
| BoxSup [ICCV '15] [11] | 62.0 | 64.6 |
| Song et al. [CVPR '19] [47] | 70.2 | - |
| BBAM [CVPR '21] [32] | 73.7 | 73.7 |
| Supervision: Image class labels | | |
| IRN [CVPR '19] [1] | 63.5 | 64.8 |
| SEAM [CVPR '20] [53] | 64.5 | 65.7 |
| BES [ECCV '20] [8] | 65.7 | 66.6 |
| Chang et al. [CVPR '20] [6] | 66.1 | 65.9 |
| RRM [AAAI '20] [61] | 66.3 | 66.5 |
| CONTA [NeurIPS '20] [62] | 66.1 | 66.7 |
| ESOL (Ours) | **69.9** | **69.3** |

Table 3: Comparison of semantic segmentation performance on PASCAL VOC 2012 validation and test set.

| Method | Sup. | val | test |
|---|---|---|---|
| SeeNet [NeurIPS '18] [19] | $\mathcal{S}$ | 63.1 | 62.8 |
| FickleNet [CVPR '19] [30] | $\mathcal{S}$ | 64.9 | 65.3 |
| CIAN [AAAI '20] [16] | $\mathcal{S}$ | 64.3 | 65.3 |
| Zhang et al. [ECCV '20] [63] | $\mathcal{S}$ | 66.6 | 66.7 |
| Fan et al. [ECCV '20] [15] | $\mathcal{S}$ | 67.2 | 66.7 |
| Sun et al. [ECCV '20] [49] | $\mathcal{S}$ | 66.2 | 66.9 |
| LIID [TPAMI '20] [40] | $\mathcal{S}$ | 66.5 | 67.5 |
| Li et al. [AAAI '21] [35] | $\mathcal{S}$ | 68.2 | 68.5 |
| Yao et al. [CVPR '21] [60] | $\mathcal{S}$ | 68.3 | 68.5 |
| ESOL (Ours) | $\mathcal{S}$ | **71.1** | **70.4** |

Table 4: Comparison of semantic segmentation performance on PASCAL VOC 2012 validation and test set using saliency maps, marked as $\mathcal{S}$.

| Method | $DT_e$ | $FL$ | $DT_s$ | $\mathcal{L}_{area}$ | seed mIoU (%) |
|---|---|---|---|---|---|
| Baseline | | | | | 48.4 |
| Expansion w/o $(DT_e + FL)$ + Shrinkage | | | ✓ | ✓ | 48.5 |
| Expansion w/o $DT_e$ + Shrinkage | | ✓ | ✓ | ✓ | 49.1 |
| Expansion w/o $FL$ + Shrinkage | ✓ | | ✓ | ✓ | 51.8 |
| Expansion + Shrinkage w/o $(DT_s + \mathcal{L}_{area})$ | ✓ | ✓ | | | 50.1 |
| Expansion + Shrinkage w/o $DT_s$ | ✓ | ✓ | | ✓ | 50.9 |
| Expansion + Shrinkage w/o $\mathcal{L}_{area}$ | ✓ | ✓ | ✓ | | 52.8 |
| Expansion + Shrinkage | ✓ | ✓ | ✓ | ✓ | 53.6 |

Table 6: Ablation studies with some components removed in our approach. $DT_e$ and $DT_s$ denote the offset learning branch in the deformable convolution layers in our Expansion and Shrinkage, respectively. $FL$ is the feature clipping operation and $\mathcal{L}_{area}$ denotes the area loss regularization. The $DT_e$ and $FL$ enforce the network pay attention to suspicious object regions as much as possible in Expansion, while the $DT_s$ and $\mathcal{L}_{area}$ enforce the network to boost the precision of the results in Shrinkage to generate high-quality localization maps.

our method achieves 71.1% and 70.4% mIoU values for PASCAL VOC 2012 validation and test set, respectively, consistently outperforming all other methods under the salient object supervision.

**MS COCO 2014 dataset:** Table 2 illustrates the segmentation performance on MS COCO 2014 compared with other methods. Our method achieves 42.6% in terms of the mIoU values on validation set, surpassing 1.2% over the IRN [1], also regarded as our baseline and outperforming the other recent competitive methods [1, 9, 53, 62] by a large margin. We further

| Method | Benchmark | GPUs | Time (h) |
|---|---|---|---|
| AdvCAM [31] | VOC2012 | 1 | 106 |
| RIB [29] | VOC2012 | 1 | 27 |
| ESOL (Ours) | VOC2012 | 1 | 10 |

Table 5: Comparison of overall pseudo-label generation time.

compare the relative improvements for comparison: CONTA reaches a 0.8% mIoU improvement compared with IRN (32.6% to 33.4%), while our method achieves 1.2% mIoU improvement (41.4% to 42.6%).

## 4.3 Ablation Studies

In this section, we conduct various ablation experiments on PASCAL VOC 2012 dataset to validate the effectiveness of each component or training scheme of our method.

### 4.3.1 Expansion Training Analysis

**Loss Maximization Controller $\alpha$:** To analyze the influence of the loss maximization controller $\alpha$ on the Expanding training sensitivity, we conduct a range from 0.001 to 0.1 for $\alpha$ as shown in Figure 6 (a). We found that too small or large $\alpha$ values degrade to attend on backgrounds extremely, showing a lower foreground recall ratio. We choose appropriate value: $\alpha = 0.01$, which balances the recall and precision of the localization maps. Visualizations are shown in the Appendix.

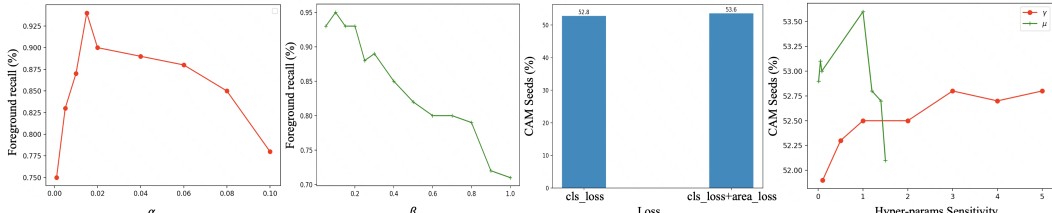

Figure 6: Ablation Studies. (a) $\alpha$ sensitivity analysis. (b) $\beta$ sensitivity analysis. (c) losses combinations. (d) $\gamma$ and $\mu$ balance analysis.

**Feature Clipping Strategy Effect:** In Expansion stage, we tend to provide high-recall foregrounds with a relatively fair chance to be sampled by Shrinkage. The feature clipping strategy is introduced after the ES and we study the impact of hand-craft threshold settings in Figure 6 (b).

### 4.3.2 Shrinkage Training Analysis

**Impact of the Loss Functions:** Although Expansion training brings high-recall foreground regions, backgrounds cannot be ignored. A classification loss and an area loss are used to optimize the network to select true foreground pixels via a loss minimization. We provide the ablative study in Figure 6 (c) to demonstrate the impact of each one and find that both of them are useful for the network training, constraint the size of the localization maps to ensure that the irrelevant backgrounds are excluded in the localization maps $\mathcal{P}_k$.

**Loss Minimization Controller $\gamma$ and $\mu$:** The sensitivity of these two hyper-parameters are performed shown in Figure 6 (d). It is observed that the performances of our approach are stable with the variation of $\gamma$ (from 0.1 to 5) and $\mu$ (from 0.01 to 1.5), *i.e.,* our method is not sensitive to such two hyper-parameters. In our experiments, the default values of $\gamma$ and $\mu$ are 1.0 and 1.0 simply.

### 4.3.3 Training and Inference Time Comparison

In this part, we make comparisons with recent SOTAs, AdvCAM [31] and RIB [29] that perform a single step to obtain the localization maps. From the Table 5, we can see that the overall generated pseudo-label time of the AdvCAM [31] and RIB [29] are much longer than ours which demonstrates that our method is more applicable in practice.

We further validate the effectiveness of each component of our methods by removing some components to perform ablative studies in Table 6. We can see that each proposed component, including deformable transformation, feature clipping operation and, training losses, enables the overall training pipeline to generate more complete and accurate initial localization maps.

## 5 Conclusion

In this paper, we explore a deformation transformation operation to address the major challenge in weakly-supervised semantic segmentation with image-level class labels. A novel training pipeline for the WSSS task, Expansion and Shrinkage, is proposed to first recover the entire target object regions as much as possible while the network is driven by an inverse image-level supervision. Then, we apply a feature Clipping operation to provide even high-recall regions for Shrinkage to sample high-precision regions, while the network is optimized by two loss functions, classification loss and area loss. Our method significantly improves the quality of the initial localization maps, exhibiting a competitive performance on the PASCAL VOC 2012 and MS COCO 2014 datasets.

**Societal implications:** This work may have the following societal impacts. Semantic Segmentation without the reliance on pixel-level annotation will save resources for research and commercial development. It is particularly useful in online network with large-scale image-tags. However, the final segmentation performance is still inferior to the fully-supervised ones and the annotations-support companies need to consider another business pattern.

## Acknowledgments

This work was supported in part by the National Natural Science Foundation of China (Grant 61871270), in part by the Shenzhen Natural Science Foundation under Grants JCYJ20200109110410133 and 20200812110350001.

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
