# Expansion and Shrinkage of Localization for Weakly-Supervised Semantic Segmentation

## A  Appendix

### A.1  Additional Analysis

#### A.1.1  Training the Expansion without Deformable Convolution Layer:

In our proposed Expansion, a deformable convolution layer with a learning offset branch is implemented with a loss maximization optimization to gradually sample the exterior object regions as much as possible. One may think a loss maximization optimization can also come to such an impact. We argue that training without introducing more sampling freedom cause no expansion at all. We conduct an additional experiment to support our argument as shown in Figure 2. We can observe that training the network with the loss maximization optimization by excluding the deformation transformation causes the network to deactivate the discriminative regions, which are activated by the original baseline model, and the expansion impact can not be observed. Besides, we obtain 40.3% mIoU scores *v.s.* 48.4% mIoU scores in terms of the initial CAM seeds on PASCAL VOC 2012 train set for the new experiment and the original baseline model, respectively. This demonstrates that the deformable convolution layer provides more sampling freedom for the network to pay more attention to other less discriminative regions, instead of dis-activating the most discriminative ones, thanks to the learning "shifts", supported by the learning offset branch inside the deformation transformation.

#### A.1.2  Loss Maximization Controller $\alpha$ Visualizations:

To better illustrate the detailed Expansion training scheme, we then provide more training visualizations to show up the results. Note that we follow the same training details as mentioned in Section 4.2 to train the Expansion model. The examples are shown in Figure 3. We can see that too small values cannot enable the learning offset fields shift to other less discriminative regions, and too large values results in large loss driven and the learning offset fields degrade to attend on background regions. Empirically, we choose a balancing setting ($\alpha = 0.01$) to achieve high-recall and less background regions.

#### A.1.3  Training Status of the Regular Convolution inside Deformable Convolution layer:

In all our experiments, we utilize the convolutional weights obtained from our baseline model to be the regular convolutional weights inside the deformable convolution layer, which is fixed during the training. We conduct an ablative study that we allow these weights to be updated as well. And we fail to capture high-recall target object regions because this degrades to the common deformable convolution layer, sharing the similar results with the experiments in Section A.1.1.

Submitted to 36th Conference on Neural Information Processing Systems (NeurIPS 2022). Do not distribute.

| Method | $DT_e$ | $FL$ | $DT_s$ | $\mathcal{L}_{area}$ | CAM seed mIoU (%) |
|---|---|---|---|---|---|
| Baseline | | | | | 48.4 |
| Expansion w/o $(DT_e + FL)$ + Shrinkage | | | ✓ | ✓ | 48.5 |
| Expansion w/o $DT_e$ + Shrinkage | | ✓ | ✓ | ✓ | 49.1 |
| Expansion w/o $FL$ + Shrinkage | ✓ | | ✓ | ✓ | 51.8 |
| Expansion + Shrinkage w/o $(DT_s + \mathcal{L}_{area})$ | ✓ | ✓ | | | 50.1 |
| Expansion + Shrinkage w/o $DT_s$ | ✓ | ✓ | | ✓ | 50.9 |
| Expansion + Shrinkage w/o $\mathcal{L}_{area}$ | ✓ | ✓ | ✓ | | 52.8 |
| Expansion + Shrinkage | ✓ | ✓ | ✓ | ✓ | 53.6 |

Table 1: Ablation studies with some components removed in our approach. $DT_e$ and $DT_s$ denote the offset learning branch in the deformable convolution layers in our Expansion and Shrinkage, respectively. $FL$ is the feature clipping operation and $\mathcal{L}_{area}$ denotes the area loss regularization. The $DT_e$ and $FL$ enforce the network pay attention to suspicious object regions as much as possible in Expansion, while the $DT_s$ and $\mathcal{L}_{area}$ enforce the network to boost the precision of the results in Shrinkage to generate high-quality localization maps.

### A.1.4   More Examples:

Figure 4 and Figure 5 present the examples of the initial localization maps for the PASCAL VOC 2012 and MS COCO 2014 dataset, respectively. Additionally, Figure 6 presents examples of the segmentation maps predicted by our final semantic segmentation model.

### A.1.5   Precision and Recall Comparisons.

Our proposed Expansion stage aims to recover the entire objects as much as possible, by sampling the exterior object regions beyond the most discriminative ones to obtain the high-recall object regions. And the proposed Shrinkage is devised to exclude the false positive regions, and thus further enhance the precision of the located object regions. As shown in Table 2, we can see that the Expansion provides high-recall CAM seeds while the Shrinkage further enhances the precision of the located regions, producing high-quality initial localization maps.

### A.1.6   More Comparisons with other SOTAs.

In this part, we make more comparisons with other SOTAs, DRS [26] nad EPS [33]. DRS [26] also adopts two-phase refinement of the localization maps, including a "Discriminative Region Suppression" classification training step and a further "Localization Map Refinement Learning" step. Noting that because of their aggressive discriminative region suppression operation behind various convolutional layers, they suffers from too much noise coming from the inevitable much background activations. They have to take advantage of the saliency cues [35] for further post-processing to achieve decent localization maps in the section "Weakly-Supervised Semantic Segmentation" in their paper, though they adopt a regressive "Localization Map Refinement Learning" to alleviate such an issue. Our approach focuses on the WSSS pipeline without introducing any other saliency prior.

For fair comparisons with DRS [26] and EPS [33], we also introduce saliency cues [35] for further post-processing on the generated PASCAL VOC 2012 pseudo groundtruths and apply the same segmentation network settings as DRS [26]. As can be seen in Table 3, when implementing stronger segmentation network settings and saliency map refinement, our final WSSS results are consistently better than DRS[26] (71.4% v.s. 71.2% mIoU on VOC2012 val and 71.8% v.s. 71.4% mIoU on VOC2012 test), noting that our previous experimental results in the original paper come from a smaller segmentation network and no additional saliency information prior refinement is performed. When compared with EPS[33], we choose the best reported results in their paper. Our new reported results are also better than EPS[33] in VOC2012 val (71.4% v.s. 71.0% mIoU). On the other hand, to show the impact of saliency map based post-processing used in DRS[26], we provide another experiment in Table 4 to compare DRS [26] and our approach in the settings of using and not using saliency cues prior, respectively.

| Method | Prec. | Recall | F1-score |
|---|---|---|---|
| IRN CVPR '19 [1] | 66.0 | 66.4 | 66.2 |
| Chang et al. CVPR '20 [2] | 61.0 | 77.2 | 68.1 |
| Expansion (Ours) | 41.1 | 90.9 | 56.6 |
| Shrinkage (Ours) | 67.1 | 79.5 | 72.7 |

Table 2: Comparison of precision (Prec.), recall, and F1-score on PASCAL VOC 2012 train images.

| Method | Segmentation Network | saliency map | val | test |
|---|---|---|---|---|
| DRS [26] | DeepLab-ASPP V3 (ResNet-101) | ✓ | 71.2% | 71.4% |
| EPS [33] | DeepLab-ASPP V3 (ResNet-101) | ✓ | 71.0% | 71.8% |
| ESOL (Ours) | DeepLab-ASPP V3 (ResNet-101) | ✓ | 71.4% | 71.8% |

Table 3: More Comparisons with other methods in terms of the final Semantic Segmentation performance.

| Method | benchmark | saliency map | seed mIoU (%) |
|---|---|---|---|
| DRS [26] | VOC2012 | ✓ | 63.6% |
| DRS [26] | VOC2012 | X | 36.0% |
| ESOL (Ours) | VOC2012 | ✓ | 67.1% |
| ESOL (Ours) | VOC2012 | X | 53.6% |

Table 4: More Comparisons with other methods in terms of the initial localization maps.

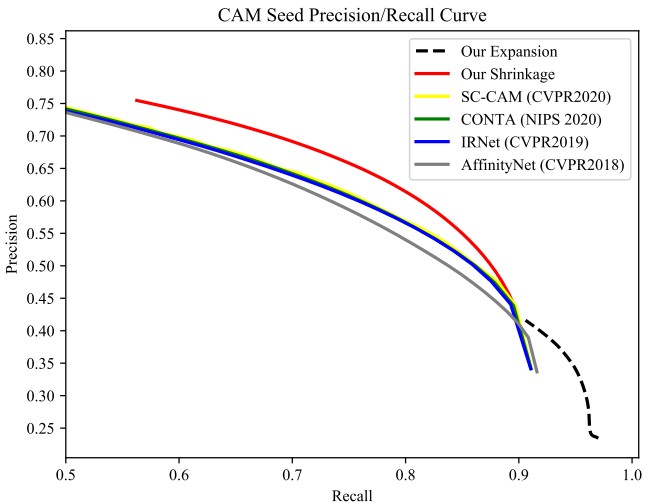

Figure 1: Precision and Recall Curve during two-phases and comparison with other SOTAs. This is obtained via setting various threshold values to calculate the corresponding precision and recall. Here we only make comparison with the methods applying refinement procedure for the fairness. Our proposed Expansion method generates high-recall results compared with other methods while the proposed Shrinkage improve the precision and obtain final high-quality localization maps.

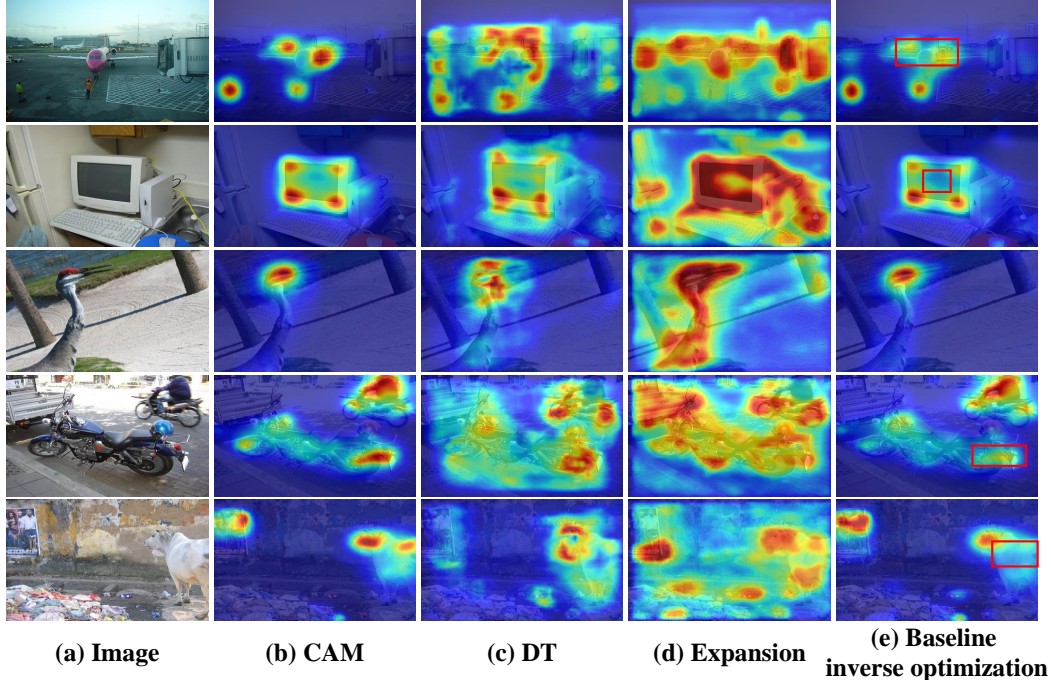

**(a) Image**     **(b) CAM**     **(c) DT**     **(d) Expansion**     **(e) Baseline inverse optimization**

Figure 2: Comparisons of localization maps on PASCAL VOC 2012 training images. (a) Input Images. (b) Original baseline CAMs. (c) DT denotes Expansion training without feature clipping strategy. (d) Expansion results. (e) Baseline applied inverse optimization only.

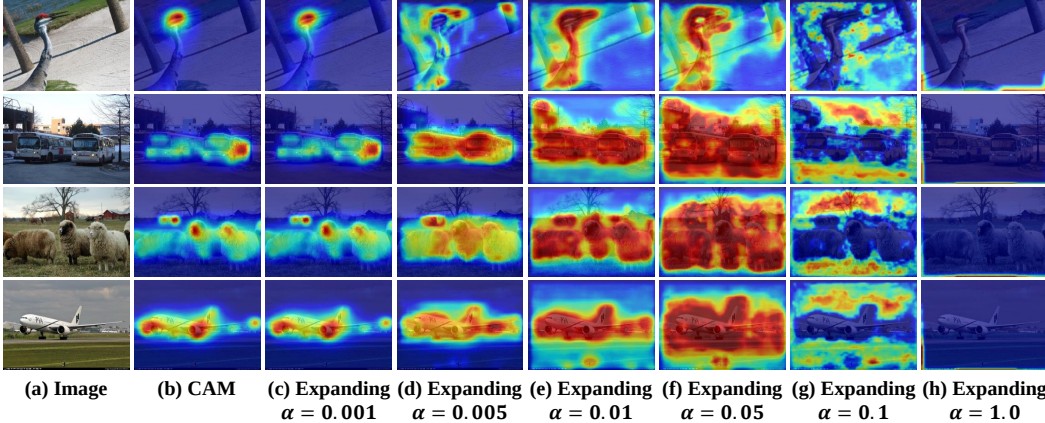

**(a) Image**   **(b) CAM**   **(c) Expanding**   **(d) Expanding**   **(e) Expanding**   **(f) Expanding**   **(g) Expanding**   **(h) Expanding**
$\alpha = 0.001$   $\alpha = 0.005$   $\alpha = 0.01$   $\alpha = 0.05$   $\alpha = 0.1$   $\alpha = 1.0$

Figure 3: Examples in terms of the sensitivity analysis of the loss maximization controller $\alpha$. (a) denotes the input images, (b) denotes original baseline CAMs, (c) $\rightarrow$ (h) means different $\alpha$ values visualization results.

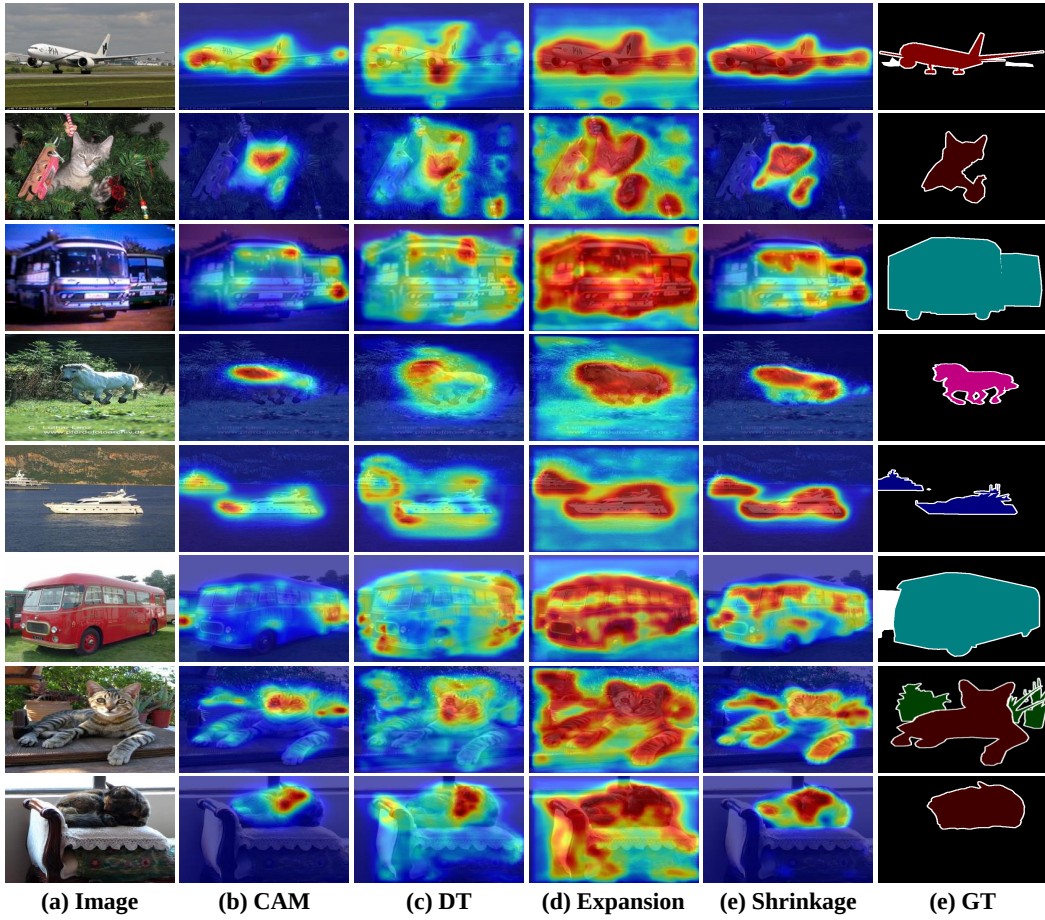

| (a) Image | (b) CAM | (c) DT | (d) Expansion | (e) Shrinkage | (e) GT |

Figure 4: Examples of localization maps on PASCAL VOC 2012 training images. (a) Input Images. (b) Original baseline CAMs. (c) DT denotes Expansion training without feature clipping strategy. (d) Expansion results. (e) Shrinkage results. (f) Ground Truth.

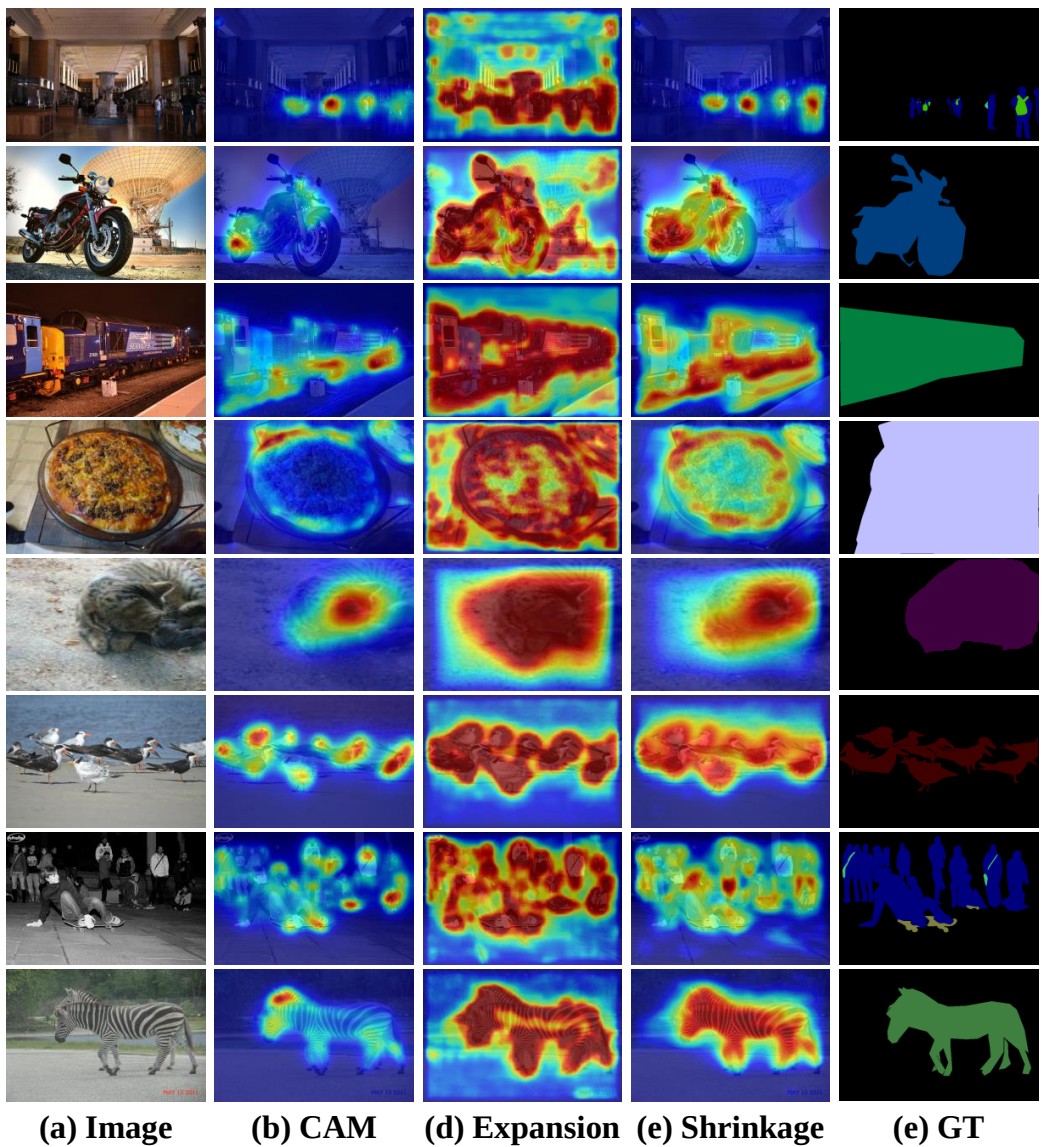

| (a) Image | (b) CAM | (d) Expansion | (e) Shrinkage | (e) GT |

Figure 5: Examples of localization maps on MS COCO 2014 training images. (a) Input Images. (b) Original baseline CAMs. (c) Expansion results. (d) Shrinkage results. (e) Ground Truth.

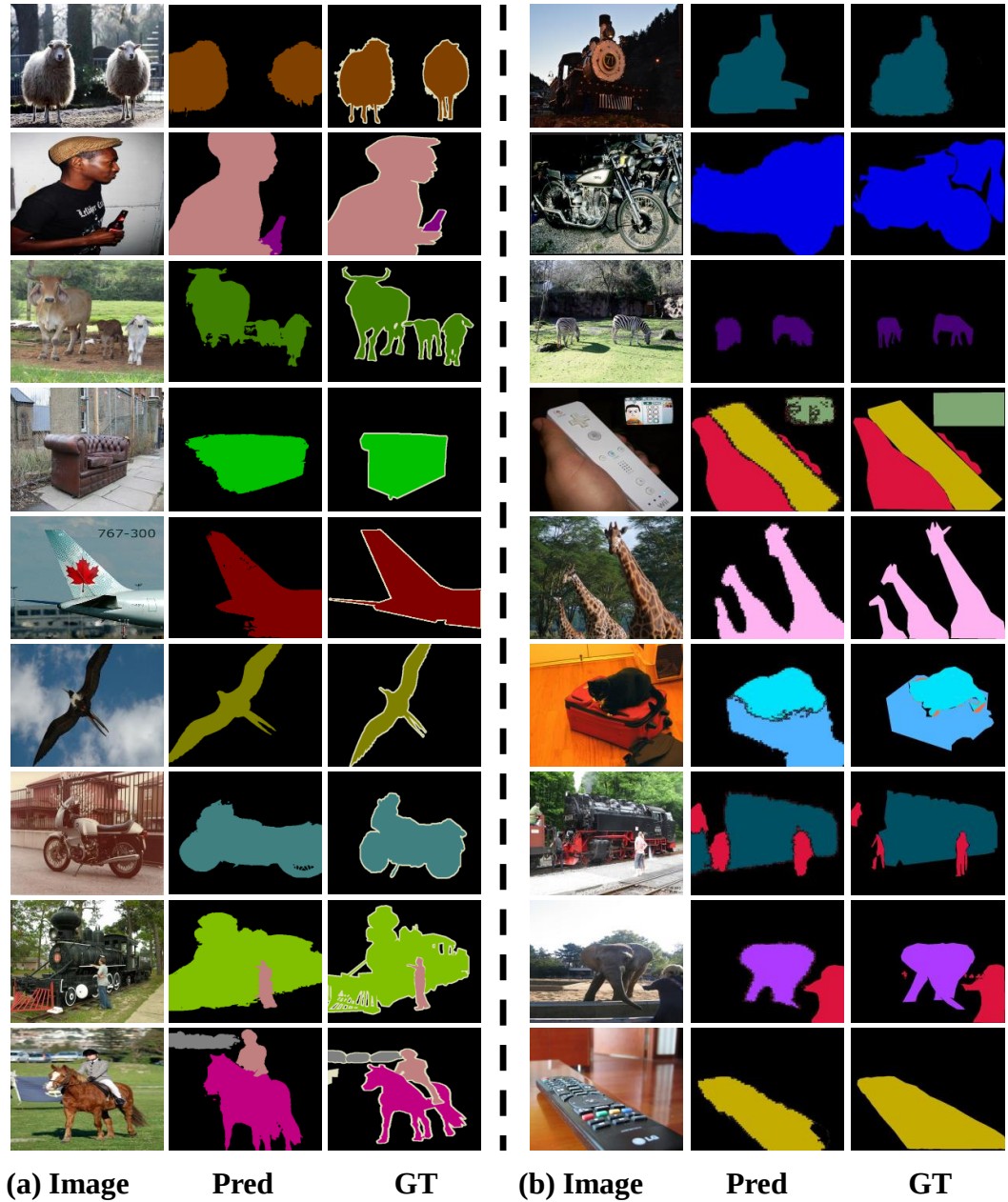

Figure 6: Examples of semantic segmentation prediction. (a) Input Images, predictions and ground truths on PASCAL VOC 2012 val set. (b) Input Images, predictions and ground truths on MS COCO 2014 val set.