# OpenReview forum: "Expansion and Shrinkage of Localization for Weakly-Supervised Semantic Segmentation"
_NeurIPS.cc/2022/Conference — NeurIPS 2022 Accept_

### Official Review · Reviewer_texX · 2022-07-10

**Rating:** 8
**Confidence:** 5
**Soundness:** 3 good
**Presentation:** 3 good
**Contribution:** 3 good

**Summary:**

This paper presents a new method for computing better class activation maps (CAMs), which have been used as crucial evidence for weakly-supervised semantic segmentation. A well-known drawback of CAM and its variants lies in that they often highlight only the small discriminative regions of an object. To alleviate such an issue, an “Expansion and Shrinkage” method is proposed, which first seeks out as much as the more positive foreground of the target object in Expansion, and then excludes the wrongly included negative regions in Shrinkage, e.g., backgrounds. In short, the proposed method embraces the offset learning of deformable convolution to introduce an expansion stage to sample out more less discriminative regions with an inverse classification loss and a shrinkage stage to filter out background regions with two losses, respectively. As a result, more object features are able to be activated so as to alleviate the partial localization issue of the CAM. The method enhances the quality of the original CAM noticeably and consequently improves the performance of a weakly-supervised semantic segmentation method relying on CAMs.

The main idea of this work is novel and interesting, and indeed useful to alleviate the well-known limitation of the CAM and its variants. Moreover, the efficacy of the proposed method has been demonstrated by extensive experiments and various ablation studies are conducted to validate the effectiveness of the proposed expansion and shrinkage stage training. This paper sheds new light on addressing the limitation of the CAM for future works with an interesting perspective.


**Questions:**

(1)Although the authors conduct various ablation studies to investigate the effectiveness of their proposed method, this reviewer still wonders how to choose the training iteration or corresponding hyper-parameters to train the classification network in the first stage, the expansion stage.

(2)The authors lack consideration if there were any possible fusion strategies to combine the generated location maps between the proposed expansion and shrinkage stage to make this paper more convincing.

(3)More insightful investigations between the proposed two stages are recommended to demonstrate, especially the changing of precision and recall during such two training stages.



**Ethics Review Area:**

["I don’t know"]

**Limitations:**

Limitations of this work are not provided in the paper, this reviewer could not find any potential negative social impact of this work. I recommend offering a discussion on the limitations in the paper, which need to be addressed in the revision.

**Strengths And Weaknesses:**

Strengths:
(1)The main idea sounds novel and interesting in that there is no previous work alleviating the limitation of the CAM and its variants through an inverse loss supervision training and feature mining via a deformation transformation.

(2)This paper is well-organized and easy to follow in most parts. And experimental details necessary to reproduce the proposed method are presented in the paper.

(3)The efficacy of the proposed method has been well demonstrated by extensive experiments on PASCAL VOC and MS-COCO, where weakly-supervised semantic segmentation based on the proposed method achieves the state-of-the-art performance. Particularly, this reviewer appreciates the CAM results of the expansion stage which seems surprising that most of the object regions are located by the proposed method with high activation response.
(4)The efficacy of hyper-parameters is investigated by extensive ablation studies. Besides the experiments in “+saliency” setups are also well-executed.

Weakness:
(1)The proposed method demands a larger number of hyper-parameters, compared to the original CAM. And also it needs more training stage to obtain the initial CAM seeds compared to other related works that also focus on improving the quality of the very initial CAM seeds.

(2)It is not explained why the final localization maps are computed only from the shrinkage stage, instead of the fusion of the proposed two training stages.

(3)Figure 4 is omitted to be referred to in the paper which causes confusion. And the final experimental setting of the kernel size of deformable convolution seems missing in the current manuscript.

(4)The authors utilize inverse loss supervision to sample out as many positive foreground regions as possible, however, it may, unfortunately, cause wrongly located target object regions in the expansion stage when there were multiple object categories present in the image. Especially the co-occurring objects appear simultaneously, e.g., dog and person. Thus, the proposed method could damage precision for the following shrinkage training, since the wrong activated features could cause confusion for the network to make the correct classification. In the same context, this reviewer would recommend analyzing, evaluating, and comparing the proposed method with other related works through precision-recall curves.

---

> ### Author Response · Authors · 2022-08-02
> **Response to Reviewer texX**
>
> Dear Reviewer texX,
>
> Thank you for appreciating our approach. We will address your comments below.
>
> **Q1. Regarding the implementation details.**
>
> A1. Thank you for your careful reading and sincere suggestion.
>
> In our Expansion phase, our motivation lies in that other less-discriminative object regions with low-response activations can get chances to be raised when introduced deformation transformation to the network with an inverse supervision signal that maximizes image-level classification loss. To achieve decent results, we carefully choose the training learning-rate and loss weight controller $\alpha$. Moreover, we have conducted ablative studies to validate our hyper-parameters choice shown in Figure 6 in our main paper and Figure 2 in our supplementary material for quantitative and qualitative demonstrations. As can be seen that too small or larger $\alpha$ will cause deactivations on target object regions or wrongly locations on too much background noise. And so as the learning-rate and training iterations did. Hope this could clarify your confusion.
>
> **Q2. Regarding the fusion results between two phases.**
>
> A2. Thank you for your constructive suggestion.
>
> Because our proposed Expansion aims to sample out as many foreground regions as possible, which is responsible for providing high-recall regions for the following Shrinkage to gradually pick out the final target object regions. Hence, the generated localization maps from the Expansion phase will ineluctably include background regions. Fusing the initial CAM seeds via max/average/min strategies is inferior to the results obtained from the final Shrinkage training. We provide further fusion experiments in Table 1 below for better illustration.
>
>                Table 1
> | Fusion    |  benchmark | train mIoU    |
> | :---:     | :----:    |  :----:        |
> | Min       |  VOC2012 train aug | 38.4% |
> | Max       |  VOC2012 train aug | 40.1% |
> | Average   |  VOC2012 train aug | 42.3% |
> | Ours |  VOC2012 train aug | **53.6%** |
>
> The fusion results show inferior performance to our approach since the high-recall results in Expansion phase along with ineluctable background regions. This also demonstrates the necessity of our proposed Shrinkage that enables the new offset learning in deformable convolution layer to gradually sample out the final target object regions via the network training.
>
> **Q3. Regarding the changing of precision and recall during training.**
>
> A3. Thank you for your careful reading and constructive suggestion.
>
> In terms of the changing of precision and recall during such two training phases, we have updated the demonstrations in our revised supplementary material in Table 2 and Figure 4 (marked red). And the results further validate the high-recall localization maps brought from our proposed Expansion and high-precision localization maps coming from our proposed Shrinkage. Thus, our proposed pipeline offers high-quality localization maps for the following stages in WSSS setting. We hope that this clarifies your confusion, but if we can be of further assistance, please let us know.
>
> Revised supplementary material: https://openreview.net/attachment?id=NM3AbzX-dq&name=supplementary_material
>
> **Q4. Regarding the Limitation and social impact discussion.**
>
> A4. We are sorry if the reviewer felt that we had not discussed the limitations or social impact of our method. In the revision, we will discuss our limitations and corresponding social impact.
>
> Limitation: In the revision, we will discuss our limitations and provide the presentation of failure cases.
>
> Social impact: Image semantic segmentation without densely pixel-level annotation will greatly save human resources for research and practical development, especially the huge number of Internet images with image-level tags, and thus has great potential in practical use. Some companies provide image annotations as a part of their services. If the dependence on image dense labels is alleviated by Weakly-supervised semantic segmentation, they may need to adjust their service style, changing their annotation style to image-tag labeling, or so on.

---

### Official Review · Reviewer_k5T6 · 2022-07-11

**Rating:** 5
**Confidence:** 4
**Soundness:** 2 fair
**Presentation:** 2 fair
**Contribution:** 3 good

**Summary:**

This paper proposes a method for weakly-supervised segmentation.

The authors refine localization maps from CAM with expansion and shrinkage steps.

The authors expand discriminative regions utilizing the deformable convolution layers in the expansion step.

The authors shrink expanded regions by minimizing the classification loss and the norm of localization maps via additional deformable convolution layers in the shrinkage step.

The authors demonstrate that refined localization maps boost the performance of weakly supervised segmentation.

**Questions:**

- I am not sure about the advantage of the proposed two-phase approach comparing one phase approach from this paper. Do the authors explain the advantage of this approach?

- DRS[3] is also a method for refinement of localization maps, and this method reports better scores than the proposed method. What is the advantage of the proposed method comparing DRS?

- The format of Table 1 in this paper seems to be similar to the format of Table 1 in the paper[4]. Why do the authors not compare the proposed method with AdvCAM[5] and RIB[4] in Table1?

- Why do the authors not list the results of RIB[4] in the Tables?

[3] Discriminative Region Suppression for Weakly-Supervised Semantic Segmentation, AAAI 2021

[4] Reducing Information Bottleneck for Weakly Supervised Semantic Segmentation, NIPS 2021

[5] Anti-adversarially manipulated attributions for weakly and semi-supervised semantic segmentation, CVPR 2021

**Limitations:**

I do not have concerns about limitations and social impacts for this paper.

**Strengths And Weaknesses:**

Pros)
- The final semantic segmentation performance is comparable with state-of-the-art methods.

- The two-phase refinement of localization maps is a novel attempt, and the authors demonstrate that the proposed two-phase steps can achieve high performance.

Cons)
- The motivation of the proposed two-phase refinement approach for localization maps is unclear.

- Important references[1][2] are missing. They report better scores on the benchmarks than the scores in this paper.

- The effectiveness of deformable convolution layers is unclear.

[1] Discriminative Region Suppression for Weakly-Supervised Semantic Segmentation, AAAI 2021

[2] Railroad is not a train: Saliency as pseudo-pixel supervision for weakly supervised semantic segmentation, CVPR 2021

---

> ### Author Response · Authors · 2022-08-02
> **Response to Reviewer k5T6 (1/3)**
>
> Dear Reviewer k5T6,
>
> Thank you for appreciating our approach and the detailed review. We will address your comments below.
>
> **Q1. Regarding the clarity of the motivation and advantage of our approach.**
>
> A1. Thank you for your detailed reading and pointing out this.
>
> The core difference between the proposed two-phase approach and the existing one-phase approaches lies in the idea of improving the recall and precision of the attended regions separately following a "divide-and-conquer" manner. We argue that it is hard to achieve a refined high-quality CAM with both better recall and precision than the initial one in a single step. This is because expanding the attended regions for higher recall deviates from excluding the false positive regions to enhance the precision in the optimization target required. Excluding false positives for higher precision usually requires the normal optimization of the model for accurate image-level classification. On the other hand, expanding the attended regions for higher recall actually demands the network to pay less attention to the pure image-level classification while focus more on scanning a wider range of the image to find suspicious regions as much as possible. Existing one-phase approaches though resort to various techniques to enlarge the search area of suspicious objects by manipulating the receptive field of the model (MDC[1],OAA[2],Ficklenet[3]), modifying the discriminative objects (Erasure[4,5,6], AdvCAM[7], DRS[8]), the search area is still constrained and the recall is limited as the model is still optimized for accurate image-level classification. Different from the one-phase approaches, the proposed two-phase approach imposes no constraint on the suspicious region discovery and even encourages it by replacing the normal optimization of image-level classification with the inverse optimization. Such a pure expansion-oriented step significantly improves the recall but meanwhile introduces many false positives. Therefore, we propose a second phase purely for boosting the precision while remaining the high recall obtained in the first step as much as possible. Moreover, our proposed two-phase method neither requires much time on learning the pseudo-labels for the training set (without iterative strategies for each image sample) nor needs extra offline saliency cues (please refer to Q3/A3&Q4/A4 for further experimental comparisons). We again appreciate and thank your review to point out this, and sincerely hope this could clarify your confusion, but if we can be of further clarity, please let us know.
>
> **Q2. Regarding the effectiveness of the deformable convolution layers.**
>
> A2. Thank you for your detailed reading and pointing out this.
>
> The proposed offset learning method in the deformable convolution layers plays different roles in our method, Expansion and Shrinkage. For the Expansion, it acts as an "expansion sampler" to locate all the possible foreground regions for the target objects, thanks to the intrinsic deformation transformation ability of deformable convolution. We have also conducted an ablative study to show the sampling effectiveness of the introduced offset learning in deformable convolution depicted in A.1.1 and shown in Figure 1 (rectangled in red in column five) in the supplementary material. As can be seen that without deformation transformation introduction, the location maps with high activated response will be deactivated compared to baseline CAM results, which can not reach our expectant high recall location maps. Therefore the introduction of the offset learning in deformable convolution layers helps to sample out more possible foreground regions and exclude wrongly activated backgrounds to narrow down to the final object regions, demonstrated visualizations shown in Figure 5 in our main paper. We provide ablative experiment results on deformable convolutional layers that compare the results of the baseline (CAM) and directly applying the classification loss maximization optimization in the network shown in Table 1 below.
>
>            Table 1
>
> | Method          |  VOC2012 train |
> | :---:           | :----:         |
> | Baseline        |  48.4%         |
> | w/o deform conv |  40.3%         |
>
> We can see that the quality of generated localization maps will degrade to a worse result without the deformation transformation in the network when optimized by the inverse loss supervision. We sincerely hope this could clarify your confusion, but if we can be of further clarity, please let us know.

---

> > ### Author Response · Authors · 2022-08-02
> > **Response to Reviewer k5T6 (2/3)**
> >
> > **Q3. Regarding the important references and corresponding experimental comparisons.**
> >
> > A3. Thank you for your detailed reading and question about more methods comparisons.
> >
> > We are sorry for missing these important references and thankful for your kind remind and we will carefully incorporate your suggestion to add these references in the revised version. DRS[8] also adopts two-phase refinement of the localization maps, including a "Discriminative Region Suppression" classification training step and a further "Localization Map Refinement Learning" step. Noting that because of their aggressive discriminative region suppression operation behind various convolutional layers, they suffers from too much noise coming from the inevitable much background activations. They have to take advantage of the saliency cues[9] for further post-processing to achieve decent localization maps in the section "Weakly-Supervised Semantic Segmentation" in their paper, though they adopt a regressive "Localization Map Refinement Learning" to alleviate such an issue. Our approach focuses on the WSSS pipeline without introducing any other saliency prior.
> >
> >
> > For fair comparisons with DRS[8] and EPS[10], we also introduce saliency cues[9] for further post-processing on the generated PASCAL VOC 2012 pseudo groundtruths and apply the same segmentation network settings as DRS[8]. As can be seen in Table 2 below, when implementing stronger segmentation network settings and saliency map refinement, our final WSSS results are consistently better than DRS[8] (71.4% *v.s.* 71.2% mIoU on VOC2012 val and 71.8% *v.s.* 71.4% mIoU on VOC2012 test), noting that our previous experimental results in the original paper come from a smaller segmentation network and no additional saliency information prior refinement is performed. When compared with EPS[10], we choose the best reported results in their paper. Our new reported results are also better than EPS[10] in VOC2012 val (71.4% *v.s.* 71.0% mIoU). On the other hand, to show the impact of saliency map based post-processing used in DRS[8], we provide another experiment in Table 3 below to compare DRS [8] and our approach in the settings of using and not using saliency cues prior, respectively.
> >
> >                              Table 2
> >
> > | Method    |  Segmentation Network | saliency map  | val   | test  |
> > | :---:     | :----:                 |  :----:      | :---: | :---: |
> > | DRS[8]    |  DeepLab-ASPP V3 (ResNet-101) |   &#10004;    |  71.2%                 |  71.4%     |
> > | EPS[10]    |  DeepLab-ASPP V1 (ResNet-101) |   &#10004;    |  71.0%                 |  71.8%    |
> > | Ours      |  DeepLab-ASPP V3 (ResNet-101) |   &#10004;    |  **71.4%**                 |   **71.8%**   |
> >
> >
> >                   Table 3
> >
> > | Method    |  benchmark | saliency map  | mIoU   |
> > | :---:     | :----:                 |  :----:      | :---: |
> > | DRS[8]    |  VOC2012 train |   &#10004;    |  63.6%              |  71.4%     |
> > | DRS[8]    |  VOC2012 train |   X    |  36.0%  |
> > | Ours      |  VOC2012 train |   &#10004;    |  **67.1%**       |   **71.8%**   |
> > | Ours      |  VOC2012 train |   X    |  **53.6%**              |   **71.8%**   |
> >
> >
> > Note that we utilize their public open source code and their pretrained checkpoints to do inference only for appropriately reproducing their reported results. We can see that the generated localization maps of DRS[8] will drastically degrade to a much worse result without extra saliency cues, from their best score 63.6% down to 36.0% in terms of the mIoU score on VOC2012 train set. This shows that DRS[8] heavily relies on extra saliency cues to exclude the background noise. On the other hand, our initial localization maps can be refined to even better quality on VOC2012 train set when using saliency maps for further post-processing. We are thankful for your important question on these comparisons and will incorporate your sincere suggestion into our final version, adding these methods (DRS[8] and EPS[10]) to our revised version and making detailed experimental results comparisons.

---

> > > ### Author Response · Authors · 2022-08-02
> > > **Response to Reviewer k5T6 (3/3)**
> > >
> > > **Q4. Regarding the comparisons with AdvCAM and RIB.**
> > >
> > > A4. Thank you for your careful reading and pointing out this.
> > >
> > > We sincerely appreciate your suggestions for making comparisons with AdvCAM[7] and RIB[11]. AdvCAM[7] proposes an anti-adversarial manner by perturbing the images along pixel gradients in the opposite direction to increase the classification score of the network and gradually identify more regions of the target object. They apply an iterative manipulation process to improve CAMs which requires several adversarial climbing iterations for each image sample. RIB[11] proposes a method to reduce the information bottleneck by devising a new loss function to replace the last double-sided saturating activation function for the classification network (e.g., sigmoid activation function). To stabilize their RIB process, they construct a batch of size $B$ by sampling random $B − 1$ samples other than $x$ at each RIB iteration, where $x$ is the fixed image. Therefore, the method of RIB[11] requires an iterative optimization process to generate the pseudo-labels one by one for the train set. We further conduct experiments to make comparisons with these methods. Our experimental environment settings are listed below:
> > >
> > > 1. Intel(R) Xeon(R) CPU E5-2690 v4 @ 2.60GHz.
> > > 2. Nvidia Titan X, 12G per card.
> > >
> > > We reproduce their results with one single GPU for fair comparison in Table 4 below in terms of overall running time comparisons of the CAM seeds improving part.
> > >
> > >                     Table 4
> > >
> > > | Method    |  benchmark | GPUs (12G)  | time (h)   |
> > > | :---:     | :----:                 |  :----:      | :---: |
> > > | AdvCAM[7]    |  VOC2012 train aug |   1    |  106  |
> > > | RIB[11]    |  VOC2012 train aug |   1    |  27                |
> > > | Ours      |  VOC2012 train aug |   1    |  **10**                 |
> > >
> > > We can see from the Table 4 that the overall generated pseudo-label time of the AdvCAM[7] and RIB[11] are much longer than ours.  Note that we only conduct comparative experiments on the benchmark VOC2012 train set with 10560 images. If a larger benchmark COCO2014 was used, their time cost would be unacceptably too long.
> > >
> > >
> > > Compared with AdvCAM[7] and RIB[11], the time of learning pseudo-labels generation of our approach is much faster which enables ours more applicable in practice, though our two-phase approach generates inferior results of initial localization maps. Our proposed pipeline lies in first searching suspicious object regions as much as possible by the introduction of the deformation transformation and inverse optimization to achieve high-recall results. And then a second phase introduced is purely for improving the precision to obtain high-quality localization maps. Our approach sheds new light on addressing the partial localization issue of the CAM for future works with an interesting perspective which is very valuable. We are very thankful for your insightful suggestions and will take these comparisons into consideration by adding these experimental results in our revised version.
> > >
> > > **[References]**
> > >
> > > [1] Revisiting dilated convolution: A simple approach for weakly-and semi-supervised semantic segmentation, CVPR2018
> > >
> > > [2] Integral Object Mining via Online Attention Accumulation, ICCV2019
> > >
> > > [3] FickleNet: Weakly and Semi-supervised Semantic Image Segmentation using Stochastic Inference, CVPR2019
> > >
> > > [4] Object region mining with adversarial erasing: A simple classification to semantic segmentation approach, CVPR2017
> > >
> > > [5] Self-erasing network for integral object attention, NIPS2018
> > >
> > > [6] Hide-and-seek: Forcing a network to be meticulous for weakly supervised object and action localization, ICCV2017
> > >
> > > [7] Anti-adversarially manipulated attributions for weakly and semi-supervised semantic segmentation, CVPR 2021
> > >
> > > [8] Discriminative Region Suppression for Weakly-Supervised Semantic Segmentation, AAAI 2021
> > >
> > > [9] A simple pooling-based design for real-time salient object detection, CVPR2019
> > >
> > > [10] Railroad is not a train: Saliency as pseudo-pixel supervision for weakly supervised semantic segmentation, CVPR 2021
> > >
> > > [11] Reducing Information Bottleneck for Weakly Supervised Semantic Segmentation, NIPS 2021

---

> > > > ### Comment · Reviewer_k5T6 · 2022-08-08
> > > > **Comments from Reviewer k5T6**
> > > >
> > > > Thanks for the detailed rebuttal comments.
> > > > I still doubt the validity of the proposed two-phase policy, but the advantage over prior works is convincing.
> > > > I agree that some prior works are complex, and the proposed approach is simple and shows high performance.
> > > > I change my rating to borderline accept.

---

> > > > > ### Author Response · Authors · 2022-08-09
> > > > > **Response to Reviewer k5T6**
> > > > >
> > > > > We are greatly thankful for your acknowledgement of our method. Our proposed two-phase method follows a "divide-and-conquer" manner that first improves the recall and then the precision of the attended regions separately. Introducing deformation transformation and inverse optimization within the two-phase policy can achieve our motivation simply and obtain high-quality pseudo-labels efficiently, which provides a new direction to address the partial localization issue for WSSS.

---

### Official Review · Reviewer_ezBz · 2022-07-11

**Rating:** 6
**Confidence:** 5
**Soundness:** 3 good
**Presentation:** 3 good
**Contribution:** 3 good

**Summary:**

This manuscript proposes a expansion and shrinkage method for weakly-supervised semantic segmentation. As the initial CAM map only focuses on discriminative regions, the authors first present the expansion module for enlarging the localization, then the shrinkage method is introduced for removing false positives. A feature clipping is also used for better shrinkage.

**Questions:**

Some symbols are not defined, thus is confusing, for example, Equ(5)

Figure 6, it is better for the authors present results with some components removed in a new table to better demonstrate their effects.

**Strengths And Weaknesses:**

Strengths :

This paper is well written and easy to follow.

As the initial CAM map only focuses on discriminative regions, the authors first present the expansion module for enlarging the localization with a deformable convolution layer. In the second stage, another deformable convolution layer is used to shrink the expanded regions. To avoid only removing the expanded regions, feature clipping is present for being shrunk with relatively fair chances.

The experimental results are good compared to state-of-the-art methods.

Ablation studies show the effectiveness of each component.

Weaknesses:

Some symbols are not defined, thus is confusing, for example, Equ(5)

Figure 6, it is better for the authors present results with some components removed in a new table to better demonstrate their effects.

---

> ### Author Response · Authors · 2022-08-02
> **Response to Reviewer ezBz**
>
> Dear Reviewer ezBz,
>
> Thank you for appreciating our approach. We will address your comments below.
>
> **Q1. Regarding the confusing symbols.**
>
> A1. Thank you for your careful reading and sincere suggestion. We have been encouraged to carefully check these details in our previous manuscript and make the following revisions in the revised version.
>
> For Equ(5), $$ \cal L_{\rm expansion} = (\rm -1.0) \cdot \alpha \cal L(\hat{\rm y}, \rm y),$$
>
> $\cal L_{\rm expansion}$ denotes the training loss function for Expansion, $\alpha$ is a hyper-parameter controlling the loss weight for updating and $\cal L(\hat{\rm y}, \rm y)$ is the commonly used multi-label classification loss function (shown in Equ(1)) while $\hat{\rm y}$ and $\rm y$ denote the model classification prediction and the given image-level labels, respectively. We have updated a revised version with the main changes marked red in the draft. We hope that this could clarify your confusion, but if we can be of further assistance, please let us know.
>
> Revised paper: https://openreview.net/pdf?id=NM3AbzX-dq
>
> **Q2. Regarding further a new table demonstration for ablation studies.**
>
> A2. Thank you for your constructive suggestion. This will help to clarify more clearly the effect of some components proposed in our paper. We have updated a new table to present results with some components removed in our newly revised supplementary material and will consider adding them in our final version. Please refer to Table 1 marked red.
>
> Revised supplementary material: https://openreview.net/attachment?id=NM3AbzX-dq&name=supplementary_material
>
> Thank you for your careful reading and insightful suggestion. We have carefully incorporated all your feedback one by one and polished the writing of the paper in the revised revision.

---

### Author Response · Authors · 2022-08-02
**General Response for all Reviewers**

Comment:

Dear all Reviewers:

We would like to thank and appreciate all the reviewers for their thoughtful and helpful comments on our submission. In this author response, we address the questions raised by the reviewers and illustrate how we will revise our submission manuscript to reflect the offered comments. Our paper tackles a well-known, yet challenging partial localization issue of CAMs. We are glad and encouraged that our paper provides a novel and interesting perspective on alleviating the well-known limitation of CAMs (R2, R3), is well written and easy to follow (R1, R3), demonstrates exhaustive and thorough investigations of the proposed idea (R1, R3), and achieves strong performance (R1, R2, R3). For specific questions of each reviewer, we answer them in the reply to each reviewer.

---

### Meta-Review · Area_Chair_cpSh · 2022-08-24

**Recommendation:** Accept
**Confidence:** Certain

**Metareview:**

All reviewers lean to accept this paper and this is a clear acceptance.

**Award:**

No

---

### Decision · Program_Chairs · 2022-09-14

Accept